# Accelerated Training on Low-Power Edge Devices

**Mohamed Aboelenien Ahmed**                                                   *mohamed.ahmed3@kit.edu*
*Karlsruhe Institute of Technology*

**Kilian Pfeiffer**                                                           *kilian.pfeiffer@kit.edu*
*Karlsruhe Institute of Technology*

**Osama Abboud**                                                       *osama.abboud@huawei.com*
*Huawei Research Center Munich*

**Ramin Khalili**                                                      *ramin.khalili@huawei.com*
*Huawei Research Center Munich*

**Heba Khdr**                                                                   *heba.khdr@kit.edu*
*Karlsruhe Institute of Technology*

**Jörg Henkel**                                                                   *henkel@kit.edu*
*Karlsruhe Institute of Technology*

**Reviewed on OpenReview:** *https://openreview.net/forum?id=cGjQ41jBEn*

## Abstract

Training on edge devices poses several challenges as these devices are generally resource-constrained, especially in terms of power. State-of-the-art techniques at the device level reduce the GPU frequency to enforce power constraints, leading to a significant increase in training time. To accelerate training, we propose to jointly adjust the system and application parameters (in our case, the GPU frequency and the batch size of the training task) while adhering to the power constraints on devices. We introduce a novel cross-layer methodology that combines predictions of batch size efficiency and device profiling to achieve the desired optimization. Our evaluation on real hardware shows that our method outperforms the current baselines that depend on state of the art techniques, reducing the training time by up to 2.3× with results very close to optimal. Our measurements also indicate a substantial reduction in the overall energy used for the training process. These gains are achieved without reduction in the performance of the trained model.

## 1 Introduction

Recent trends are moving toward the deployment of deep learning models on edge devices due to the need for real-time decision-making, reduced communication cost, and privacy compared to offloading computation to cloud-based servers Wang et al. (2018); He et al. (2018); Chen & Ran (2019); Ganesh et al. (2022). Low-latency inference is essential in applications such as autonomous vehicles, robotics, surveillance, and smart agriculture. To meet these requirements, edge devices are equipped with compact GPUs optimized for computer vision and deep learning inference tasks Maghazeh et al. (2013); Barba-Guaman et al. (2020); Saddik et al. (2021); Fernández-Sanjurjo et al. (2021); Zhang et al. (2018); Ganesh et al. (2022). Such devices are often power constrained, running on batteries with limited power outputs.

Many applications on edge devices involve sensitive or private information that cannot be shared due to regulatory or ethical concerns, making it challenging to maintain accurate models without access to this data. Training on the edge offers adaptability to the data on these devices without transmitting it, reducing exposure risk. Transfer learning is thereby applied for on-device training, where a model is pre-trained with

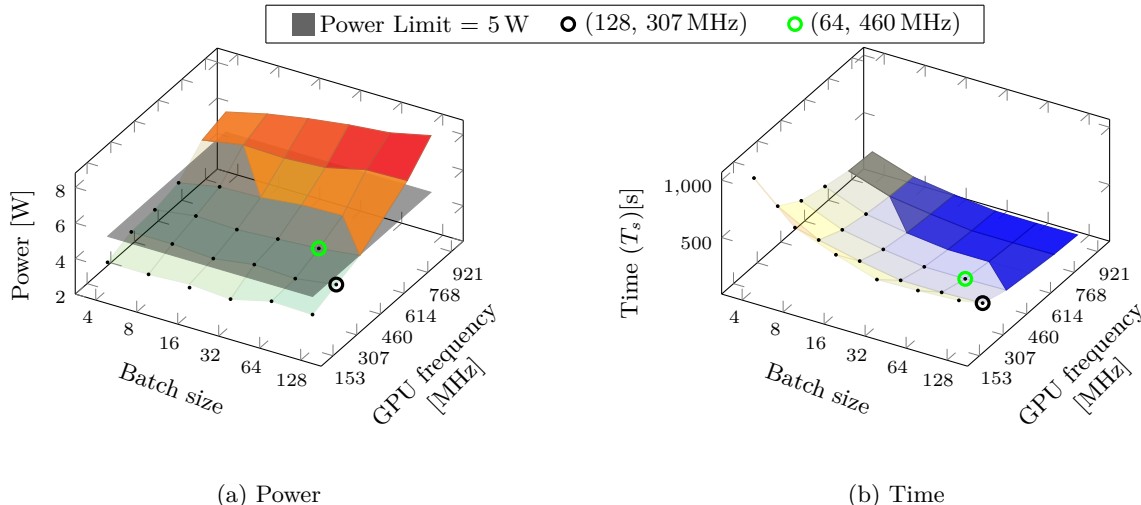

(a) Power

(b) Time

Figure 1: Peak power and time for a single training pass over a set of samples across different batch size and GPU frequency combinations. The gray plane represents the power limit in the left figure, and the black dots in both figures are the feasible combinations that can be utilized under that constraint. The black circle represents the operating point with maximum feasible frequencies for the batch sizes of 128, which will be selected by the state-of-the-art techniques. The green circle represents an operating point at batch size 64, that could be selected when the frequency and the batch size are jointly selected to accelerate training under the power constraint. Selecting this operating point accelerates training by 31.9% (see Time curves).

a large (public) dataset and fine-tuned on-device on the target domain Cai et al. (2020); Chiang et al. (2023); Lin et al. (2022); Yang et al. (2023). While this offers less training time compared to training from scratch, transfer learning is still a computationally demanding task utilizing higher power compared to inference.

To accelerate training, usually large batch sizes for mini-batch gradient descent are used. However, increasing the batch size increases the power consumption which is limited at edge devices. To enforce the power constraint, the circuit-level control will select the frequency level that satisfies this constraint considering the worst-case computation scenario. Intuitively, this will decelerate training: in the state of the art, *the system-level and application-level (training) parameters are set independently from each other*, and *this is sub-optimal* as will be demonstrated in the following example.

Figure 1 shows the power and time required to train a ResNet18 He et al. (2016) model on Nvidia Jetson Nano with 4096 of CIFAR10's training samples, while using different combinations of the GPU's frequency and batch size. Following the state of the art, and for a power constraint of 5 W, the system would select the GPU's frequency $f = 307$ MHz and the batch size $b = 128$[1]. When we look at the training time, this selection is sub-optimal. In particular, we could choose a smaller batch size ($b = 64$), allowing the GPU to operate at a higher frequency ($f = 460$ MHz), and thus reducing the training time by 31.9%. *This shows that to accelerate the training on low-power edge devices, both the GPU frequency (a system parameter) and the batch size (an application parameter) should be jointly selected.*

Motivated by this observation, we introduce, for the first time, a cross-layer approach that simultaneously optimizes both the GPU frequency and the batch size used for training, aiming to accelerate the process while adhering to device power restrictions. Note that the example in Fig. 1 illustrates the impact of batch size on *a single training pass*. However, modifying the batch size in training (in contrast to the inference process) influences the number of iterations needed to reach the targeted accuracy, and consequently *the overall training time*. This presents an additional dimension to the problem, which we will consider in our approach as detailed in Section 3. Furthermore, our cross-layer optimization is capable of leveraging operating points within the design space that previous works have not considered. This approach facilitates a marked decrease in the necessary training time to accuracy.

---

[1]We select the maximum batch size which is allowed by the memory at the device.

Our solution incorporates offline profiling of Neural Network (NN) training on the device, where we assess power and time demands for different combinations of batch sizes and GPU frequencies. Furthermore, we estimate how batch size correlates to the number of iterations required to attain a desired accuracy by training on a proxy dataset on a server. By merging these insights, we identify the combination that minimizes overall training time on the device while respecting its specified power constraints.

In summary, we make the following novel contributions:

- We motivate and demonstrate the importance of the joint optimization of GPU frequency and batch size to accelerate on-device training under a power constraint, as shown across representative models (CNNs and Transformers) and tasks (image classification and next character prediction).

- We introduce a new cross-layer approach that performs the discussed joint optimization, taking into account the effect of batch size on the overall training time, aiming to accelerate on-device training while complying with the device's power limitations.

- Our evaluation on real hardware under different power constraints for image classification task on SVHN Netzer et al. (2011) and CINIC Darlow et al. (2018) datasets, using models ResNet18, MobileNetV2 Sandler et al. (2018), and EfficientVIT Liu et al. (2023)[2], and next character prediction task using a transformer network shows a significant improvement in the training speed by an average gain of $1.5\times$ and peak of $2.3\times$ compared to baselines. We observe a reduction in the total energy used for the training at the device.

- We provide a comprehensive sensitivity analysis that demonstrates the robustness of our solution towards the proxy dataset selection over five image classification datasets and three next character prediction datasets, validating the practicality of our approach and ensuring its effectiveness.

## 2 Background and Related Work

This section discusses the control parameters (i.e., GPU frequency and batch size) used in this work and the state-of-the-art techniques. We do not consider training on the CPU for the GPU-equipped devices, as the GPU is an order of magnitude faster (and energy efficient) while consuming nearly the same average power (see Appendix A.1 for some related discussion).

### 2.1 Voltage Frequency Scaling (VFS)

VFS is a system-level technique used for power management in processors Cochran et al. (2011); Lee et al. (2011). It adjusts the voltage and frequency at runtime, providing a trade-off between performance and energy efficiency Le Sueur & Heiser (2010); Guerreiro et al. (2019). Reducing each of the processor's frequency $f$ and voltage $V$ values can result in a cubic reduction in the dynamic power, i.e., $P_{\text{dynamic}} \approx fV^2$. Within a device, the chip manufacturer provides a discrete set of frequencies to operate with, where $f_i$, $i \in \{1, 2, \ldots, \max\}$. The voltage is a function of the operating frequency $f$, and will be updated automatically for any given frequency. To enforce power constraints on processors, the industry standard technique is to have an upper bound $f$ (and $V$) for the device to use under a power limit. VFS is also applied to the CPU's frequency, but this is beyond the scope of our work as we train on the GPUs.

VFS has been studied in Nabavinejad et al. (2019); Liu & Karanth (2021) for improving performance and energy efficiency for inference only. Tang et al. (2019) conducted a study on the impact of GPU VFS on performance and energy for both training and inference. However, the study mainly focused on the computational perspective without addressing accuracy and training speed till convergence. In contrast, we show how changing the frequency and batch size due to power limits, accompanied by the difference in iterations to reach accuracy, can lead to different optimal configurations in training.

---

[2]The selected models and tasks align with existing literature for training on edge devices Ponzina et al. (2023); Choi & Sobelman (2022); Jia et al. (2024); Pfeiffer et al. (2023), and Jetson Nano and TX2NX represent typical edge GPU platforms.

## 2.2   Batch Size

One of the state-of-the-art techniques for training NNs is the minibatch gradient descent, where gradients are computed using the samples drawn in mini-batch to approximate the overall gradient, enabling iterative updates of the model parameters. The latency for processing a batch is influenced by the GPU's ability to parallelize computations and its memory bandwidth. Increasing the batch size generally decreases the latency of processing data samples by parallelizing more computations on multiple processing units. Nevertheless, this increases the number of computational operations on the GPU, and hence the power consumption.

The impact of batch size on accuracy and convergence speed is explored from multiple aspects. Goyal et al. (2017); Krizhevsky (2014); Zhang et al. (2019) studied the interdependencies between batch size and other hyperparameters such as learning rate, weight decay, and optimizers. Smith et al. (2018) proposed to increase the batch size during training rather than decreasing the learning rate to leverage the regularization effect employed by larger batch sizes. McCandlish et al. (2018) proposed a gradient noise scale metric to predict the most efficient batch size that maximizes the throughput in the next training step. This approach is tailored for distributed learning, where a batch of data is split across multiple devices. These aforementioned works have not considered the power efficiency of batching on the device. In contrast, the works in Nabavinejad et al. (2021; 2022) consider the impact of the batch size on power consumption of inference operations. In particular, a binary search approach is employed in Nabavinejad et al. (2022) to find an appropriate batch size that accelerates inference and then the GPU frequency is adjusted accordingly to satisfy power constraints on GPU servers. *Most importantly, this approach only considers inference where the statistical impact of the batch size does not exist.* You et al. (2023) aimed at optimizing the energy efficiency of periodic training jobs (continuously re-executed for incoming data flow) on powerful GPU clusters. To achieve their optimization (which is different from the one we target in this paper), they propose to adjust the batch size, set a power limit on the server, and depend on circuit-level control to select the frequency that satisfies that power limit. In particular, they obtain Pareto-optimal combinations of batch sizes and power limits that optimize for energy and performance through profiling the whole training job on the server, since this job will be repeated for new data flows. However, this solution can be applied on powerful GPU clusters and not on edge devices with limited resources.

*In summary, none of the state of the art has addressed the joint selection of the batch size and GPU frequency to accelerate training at edge devices under power constraints.*

## 3   Problem Statement

We consider the following scenario: for a specific training task, an edge device requests a pre-trained NN model $M$ with its weights $\theta$ from a server in order to fine-tune it on local data $D$ till reaching a given accuracy threshold. Importantly, the edge device has a power constraint $P_{\max}$, which should not be exceeded during the training process. Our goal in this paper is to *minimize the training (fine-tuning) time at edge devices under their given power constraints.*

We introduce $T_s$ as the training time required to apply training using a fixed number of samples $s$. As shown in Fig. 1, the joint selection of $b$ and $f$ will help reduce $T_s$ under a power constraint. However, the ultimate optimization goal is to minimize the total training time required to reach a target accuracy, which we label $TT_{\text{acc}}$. A set of parameters, i.e., frequency and batch size $(f, b)$, that are optimal for training a fixed number of samples $(T_s)$ might not be necessarily optimal for the training to accuracy $(TT_{\text{acc}})$.

We display in Fig. 2 the training time to reach an accuracy of 78% ($TT_{\text{acc}}$) for ResNet18 using two batch sizes of $b_1 = 8$ and $b_2 = 32$ under three different power constraints (i.e., $P_1 = 4.5\,\text{W}$, $P_2 = 5\,\text{W}$, and $P_3 = 7\,\text{W}$). We notice that for the three power constraints, selecting $b_1$ allows to utilize a higher frequency than $b_2$. For each batch size, we select the highest frequency that satisfies the power constraint, and measure $T_s$ and $TT_{\text{acc}}$. We observe that using $b_2$ (the higher batch size) always leads to a lower $T_s$. However, selecting the same batch size over $b_1$ leads to a longer $TT_{\text{acc}}$ for $P_1$ and $P_2$, and shorter $TT_{\text{acc}}$ for $P_3$.

This shows the complexity of the targeted problem. In particular, $TT_{\text{acc}}$ does not only depend on $T_s$, but it also depends on the number of times of processing $s$ to reach target accuracy $(N_{s_{\text{acc}}})$. In this example, $N_{s_{\text{acc}}}$

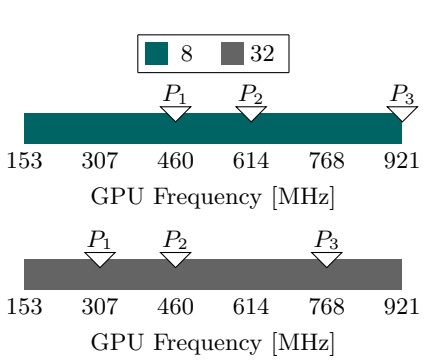
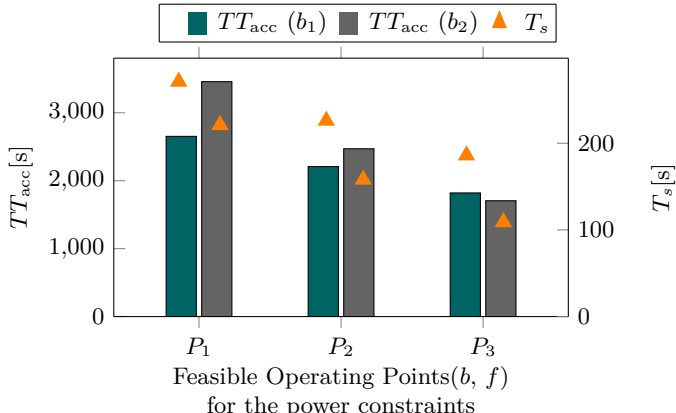

Figure 2: The training time on fixed number of samples $T_s$ and the total training time $TT_{\mathrm{acc}}$ to reach an accuracy threshold of 78% using two batch sizes 8 and 32, while considering the maximum feasible GPU frequencies under three power constraints; $P_1 = 4.5\,\mathrm{W}$, $P_2 = 5\,\mathrm{W}$, and $P_3 = 7\,\mathrm{W}$. We observe that for $P_1$ and $P_2$, selecting $b = 8$ will lead to lower $TT_{\mathrm{acc}}$, while for $P_3$ selecting $b = 32$ is better. This is in contrast with our observation for $T_s$, where selecting $b = 32$ is the best option in all cases.

for $b_2$ is equal to 15 while $N_{s_{\mathrm{acc}}}$ for $b_1$ is equal to 10. These values and the effect of power constraint on the feasible frequency highly influence the optimal batch size to minimize $TT_{\mathrm{acc}}$. In summary, there is no clear indication on how to select the optimal operating points $(f, b)$ to achieve the target goal. We formulate our optimization problem as follows:

$$\min_{b \in \mathcal{B}, f \in \mathcal{F}} \quad TT_{\mathrm{acc}}(b, f, M, D) \tag{1}$$
$$\text{subject to} \quad P(b, f, M) \leq P_{\max}$$

where $\mathcal{B}$ is the set of feasible batch sizes, $\mathcal{F}$ is the set of available GPU's frequencies, and $P(b, f, M)$ is the required power to training $M$ using $b$ and $f$. We rewrite $TT_{\mathrm{acc}}$ as the multiplication of $T_s$ and $N_{s_{\mathrm{acc}}}$, we thus have:

$$TT_{\mathrm{acc}}(b, f, M, D) = T_s(b, f, M) \times N_{s_{\mathrm{acc}}}(b, M, D) \tag{2}$$

$s$ is selected, s.t. $b_{\max} \leq s \leq |D|$, where $b_{\max}$ is the largest batch size that can fit into the memory of the devices. This detached formulation enables our proposed optimization method, presented in Section 4. In particular, the first factor $T_s$ does not depend on the training data $D$, nor on the accuracy threshold. The second factor $N_{s_{\mathrm{acc}}}$ is independent of the GPU frequency of the device.

## 4 Power-Aware Training

We propose an optimization approach that co-selects $b$ and $f$ to minimize $TT_{\mathrm{acc}}$ on an edge device under $P_{\max}$. Following the problem split that we propose in Equation (2), our solution consists of two main parts. The first one considers device specifics, and involves measuring time and power for training a given model $M$, i.e., $T_s(b, f, M)$ and $P(b, f, M)$ for every $b \in \mathcal{B}$ and $f \in \mathcal{F}$. The second part is responsible for estimating the efficiency of batch sizes, i.e., their impact on $N_{s_{\mathrm{acc}}}$. This part does not depend on device characteristics and it is computationally expensive, therefore we consider it to be performed on the server that is responsible for pre-training and sending of the model.

The workflow for our approach is as follows: The device first sends a request to the server for a pre-trained neural network model ($M$) with a specific architecture and input size. The server responds with the requested model and its pre-trained weights. Our proposed power-aware training then begins. The device profiles the model in terms of time and power as will be described in Section 4.1, while the server estimates $N_{s_{\mathrm{acc}}}$ for various batch sizes, as will be discussed in Section 4.2, and sends these estimations to the device. Based on the profiling and estimations, the device selects the best combination of batch size and frequency to minimize $TT_{\mathrm{acc}}$. An overview is presented in Figure 3.

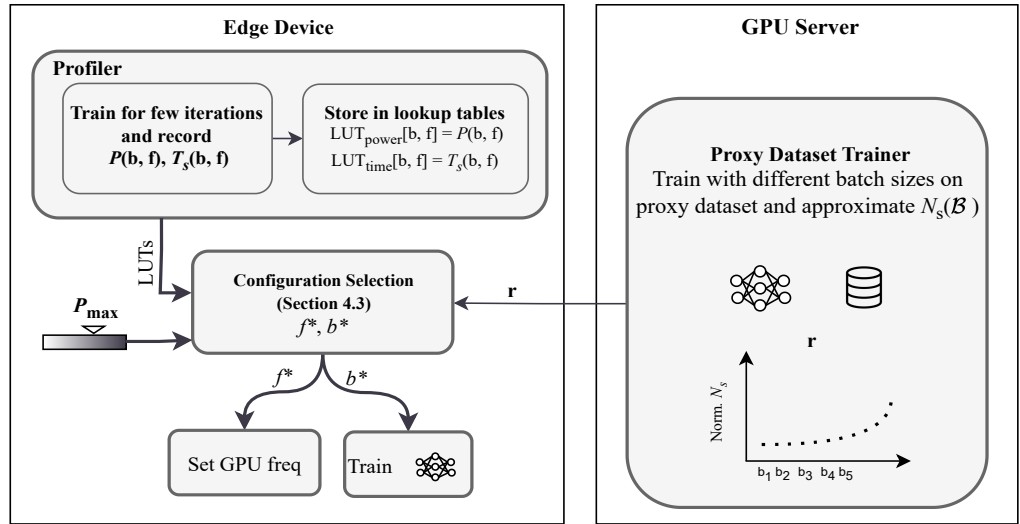

Figure 3: Overview of our proposed cross-layer approach that accelerate training under power constraint though the joint selection of batch size $b$ and GPU frequency $f$.

## 4.1 Profiling

The power and time for training on a device also depend on the NN's architecture $M$, in addition to the frequency and batch size. Besides, changing the input shape, such as the image size in vision tasks, can lead to different time and power requirements even when using the same NN architecture. Furthermore, the time and power are independent of the actual training data. Therefore, profiling $T_s(b, f, M)$ and $P(b, f, M)$ given the task's input shape can be performed before full data acquisition, as long as the input dimensions of the data samples are known.

Processing a few mini-batches is adequate to obtain accurate profiling. For each batch size and frequency combination, we set the GPU frequency to $f$ and process a few mini-batches $m$ of size $b$. The power sensor values are monitored to extract the peak power. Furthermore, the processing time is recorded, and then the average processing time for a mini-batch is calculated. This average processing time is then scaled to the processing time for $s$ samples, denoted as $T_s$. $T_s(b, f, M)$ and $P(b, f, M)$ are then stored in two lookup tables, denoted as $\mathrm{LUT}_{\mathrm{Time}}^{M}$ and $\mathrm{LUT}_{\mathrm{Power}}^{M}$.

The proposed profiling strategy considers hardware and NN structures. Also, the time and power values are not affected by the network weight updates; thus, profiling is applicable before receiving pre-trained weights if $M$ is available at the device and is needed to be perdomed only once. In a future and more practical setting, we could assume such profiling to be provided by the device manufacturer and for a pre-defined set of tasks and models. This can be further supported by periodic validation of current values with those recorded in the LUTs. If a deviation is detected, specifically relative performance change between operating points (e.g., due to memory or cache contention), a re-profiling procedure is initiated to accommodate the change, requiring only a few batches per operating point, that is much cheaper than full training. This process can be further optimized by interpolating the deviations across operating points, thereby enabling efficient updates to the LUTs with minimal computational overhead.

Furthermore, if $P_{max}$ is known at the device, the profiling can be performed more efficiently as follows. Given that $\mathcal{B}$ and $\mathcal{F}$ are sorted, and power consumption increases with both $b$ and $f$, we begin by profiling the largest $b$ with the minimum $f$, incrementing $f$ until the maximum feasible value under $P_{max}$ is reached. Next, we move to the second largest $b$, starting from the highest feasible frequency found for the previous batch size, and repeat the process until the smallest batch size is profiled. This profiling is equivalent at most to 1.9% and 6% of the training times performed on ResNet18 and MobileNetV2 in Section 5.2.

## 4.2 Estimation of $N_{s_{\text{acc}}}$ for Batch Sizes

As discussed in Section 3, batch sizes have different efficiency in terms of $N_{s_{\text{acc}}}(b_i, M, D)$. Estimating $N_{s_{\text{acc}}}(b_i, M, D)$ for every $b_i \in \mathcal{B}$ depends also on the training data that is only available at the device and $M$'s pre-trained weights. To solve Equation (1), the exact number of samples processed to reach target accuracy for each batch size can be replaced with the relative ratio between batch size $r_{b_i}$ (i.e., normalized to the maximum $N_{s_{\text{acc}}}$) as follows:

$$r_{b_i} = \frac{N_{s_{\text{acc}}}(b_i, M, D)}{\max_{b \in \mathcal{B}} N_{s_{\text{acc}}}(b, M, D)} \tag{3}$$

With this simplification, we adjust our focus to estimate the relation vector between batch sizes such that $\mathbf{r} = (r_{b_1}, r_{b_2}, \ldots, r_{b_{\max}})$, where $r_{b_i} \in (0, 1]$. However, this is still a complex task to solve given the non-linear training dynamics of deep learning; especially as the convergence speed of every batch size changes across training, making it impossible to estimate with few probes of multiple batch sizes given the training state. Obviously, training till convergence for multiple batch sizes is computationally-expensive task and cannot be conducted on the device. These all make on-device estimation of $\mathbf{r}$ inaccurate, if not infeasible.

We thus propose to estimate $\mathbf{r}$ on a powerful GPU server. Particularly, $M$ is trained on the server with multiple batch sizes until convergence (reaching the target accuracy) using a proxy dataset $D_S$ since the server does not have access to $D$. Each batch size run starts from the same pre-trained weights and yields its corresponding $N_{s_{\text{acc}}}$ on $D_S$. This systematic exploration allows us to comprehensively assess the long-horizon impact of the batch size on the model's convergence while leveraging the computational capabilities of the server, along with the datasets available on it and augmentation techniques. Since $M$, which would be used on the edge device, is already designed for a specific task type and pre-trained on a public dataset, a proxy dataset $D_S$ should share the same task type (e.g., image object classification) and similar input shapes. Training network $M$ on $D_S$, despite having different objectives, allows us to estimate the relationship between batch sizes and their relative examples to accuracy on $D$. Thus, we can finally have a mapping such that $\mathbf{r}_{D_S} \approx \mathbf{r}_D$. By estimating the batch size relation vector $\mathbf{r}$ on the server, any edge device aiming to train $M$ to utilize this vector. In Section 5.2, we provide evaluation for two different devices, namely Nvidia Jetson Nano and Nvidia Jetson TX2NX, utilizing $\mathbf{r}$.

## 4.3 Batch Size and Frequency Selection

The device profiling and estimation of batch size efficiently are performed in an offline manner and at the design time. In contrast, the configuration selection is performed at runtime, as described below. Given a power constraint $P_{\max}$, and the power measurements stored for training a specific $M$ in $\text{LUT}^M_{\text{Power}}$, we construct a set of feasible combinations $C$ consisting of every feasible batch size with its corresponding highest (and fastest) frequency satisfying $P_{\max}$ as follows:

$$
\begin{aligned}
C &\leftarrow \{(i, j_i))|i \in [1, \ldots, |\mathcal{B}|]\}, \\
j_i &\leftarrow \max\{j|j \in [1, \ldots, |\mathcal{F}|], \text{LUT}^M_{\text{Power}}(i, j) < P_{\max}\}
\end{aligned} \tag{4}
$$

The processing time for the feasible combinations is then extracted from $\text{LUT}^M_{\text{Time}}$. Following this, an approximate training time is computed by multiplying the time for every $b_i$ (and frequency) by the corresponding $\mathbf{r_i}$ element from the relation vector $\mathbf{r}$ (i.e., estimated at design time). The configuration $(b^*, f^*)$ that minimizes the approximate total training time is selected. Finally, we set the GPU frequency to $f^*$ and then start the training using batch size $b^*$. We provide the configuration selection in Algorithm 1. The selection part is $\mathcal{O}(n^2)$ in the worse case scenario ($\mathcal{O}(|\mathcal{B}| \times |\mathcal{F}|)$). Since $\mathcal{B}$ and $\mathcal{F}$ are sorted, and for each $b$, the power/time over $\mathcal{F}$ are also sorted, the selection reduces to $\mathcal{O}(n \log n)$ via binary search and is negligible to training time.

## 5 Results

In this section, we evaluate the training time and energy consumption for finetuning vision and text tasks on Nvidia Jetson Nano and TX2NX. Additionally, we assess the effectiveness of our approach by conducting a sensitivity analysis on the selection of the proxy dataset.

---

**Algorithm 1** Batch size and GPU frequency selection

---

**Require:** Power Limit $P_{\max}$, Neural Network $M$, List of feasible batch sizes $\mathcal{B}$, List of GPU frequencies $\mathcal{F}$, Dataset at server $D_S$

1: **Server:**
2: $\mathbf{r} \leftarrow \text{Proxy}(\mathcal{B}, M, D_S)$ ▷ See section (Section 4.2)
3:
4: **Device:**
5: Let $\text{LUT}^M_{\text{Time}} \in \mathbb{R}^{|\mathcal{B}| \times |\mathcal{F}|}$
6: Let $\text{LUT}^M_{\text{Power}} \in \mathbb{R}^{|\mathcal{B}| \times |\mathcal{F}|}$
7: **for** $i \in 0, 1, ..., |\mathcal{B}|$ **do**
8:      $b \leftarrow \mathcal{B}[i]$
9:      **for** $j \in 0, 1, ..., |\mathcal{F}|$ **do**
10:          $f \leftarrow \mathcal{F}[j]$
11:          time, power $\leftarrow \text{Profile}(b, f, M)$ ▷ On Device (Section 4.1)
12:          $\text{LUT}^M_{\text{Time}}[i, j] \leftarrow \text{time}$
13:          $\text{LUT}^M_{\text{Power}}[i, j] \leftarrow \text{power}$
14: *// On-device Configuration Selection:*
15: $TT_{\text{acc}} = \text{LUT}^M_{\text{Time}} \times \mathbf{r}$ ▷ Element-wise multiplication
16: $C \leftarrow \text{GetCombinations}(\mathcal{B}, \mathcal{F}, \text{LUT}^M_{\text{Power}}, P_{\max})$ ▷ Based on Equation (4)
17: $TT^p_{\text{acc}} \leftarrow TT_{\text{acc}}[C]$
18: $(b\_\text{idx}, f\_\text{idx}) \leftarrow \mathbf{argmin}(TT^p_{\text{acc}})$
19: $b^*, f^* \leftarrow \mathcal{B}[b\_\text{idx}], \mathcal{F}[f\_\text{idx}]$ ▷ Selected batch size and GPU frequency

---

## 5.1 Experiment Setup

**Datasets and models:** We evaluate our approach for the image object classification and next-word prediction tasks. For image classification, a model is pre-trained with the full CIFAR100 Krizhevsky et al. (2009) and to be trained on subsets (i.e., quarter) of SVHN and CINIC datasets on the device. We use a subset of CIFAR10 Krizhevsky et al. (2009) as the proxy dataset on the server. For all datasets, the input images are of size $3 \times 32 \times 32$. Each dataset (subset) is divided into training and testing sets with an 80/20 split, with target accuracy evaluated on the test split. We evaluate ResNet18, MobileNetV2 and EfficientViT-(M1) models, trained with Adam optimizer Kingma & Ba (2015). A summary of the experimental settings is provided in Table 1. For the next character prediction task, we evaluate a 6 layer transformers with 6 attention heads per attention block, 256 embedding dimensions, and a sequence length of 64. We use AdamW Loshchilov & Hutter (2017) as an optimizer. We pre-train the model on WikiText-2 dataset Merity et al. (2016), utilize tiny shakespeare Karpathy (2015) as a proxy dataset, and train it on some Jane Austin and Charles dickens novels[3]. We use 90% of a dataset for training and the rest for testing. For all datasets, we fix the vocabulary to include only words with English letters, digits, punctuation, spaces, and new lines. We set the target character level accuracy for Austin and Dickens at 62% and 61%, respectively.

**Batch size and learning rate:** The choice of appropriate learning rate and batch size are often intertwined, as they impact each other's effectiveness. Larger batch sizes provide more stable gradient estimates, potentially permitting the use of higher learning rates. Therefore, to preserve the performance of deep models with different batch sizes, we apply learning rate scaling (i.e., square root scaling Krizhevsky (2014) for Adam and AdamW). For ResNet18, the batch sizes ranged from 4 to 128, consisting exclusively of powers of two. The initial learning rate of $5 \times 10^{-4}$ is used for the largest batch size of 128 (with learning rates scaled for other batch sizes). The same setup was also applied to MobileNetV2 and EfficientViT; however, the batch size of 128 was omitted due to memory constraints. For transformers, we similarly consider batch sizes of 4 to 128 with a learning rate of $1 \times 10^{-3}$ for the batch size of 128.

---

[3]We used the text for works of Jane Austin's from nltk package Bird et al. (2009) and downloaded Dickens works from project Gutenburg. More details are provided in Appendix A.3

Table 1: Experimental setup for image classification datasets and models used.

| Dataset | Dataset size | Target Acc. | | |
|---------|-------------|-------------|--------------|--------------|
| | | ResNet18 | MobileNetV2 | EfficientViT |
| Cifar10 | 12500 | 87% | 84.5% | 84% |
| SVHN | 18315 | 92% | 91.5% | 90% |
| CINIC | 22500 | 76% | 74% | 72% |

**Hardware and power limits:** We evaluate our method on Nvidia Jetson Nano with $4\,\mathrm{GB}$ memory on three scenarios. In the first and second scenarios, the power limits at the device are set to $P_{\max}^1 = 4.5\,\mathrm{W}$ and $P_{\max}^2 = 7\,\mathrm{W}$. In the third, the device operates without any power limits, denoted as N/A. To show that our solution and our results are not device specific, we also provide evaluation on another device (i.e., Nvidia Jetson TX2NX). We use PyTorch 1.10 Paszke et al. (2019) for Jetson Nano and TX2NX. For pretraining and proxy datasets' training we use NVIDIA A6000 GPU with Pytorch 2.1.

**Comparison baselines:** To highlight the impact of joint selection of GPU frequency and batch size, we compare our approach to the following baselines that depend on state of the art techniques:

- **Baseline 1:** The state of practice is to use the largest $b$ that can fit into memory Goyal et al. (2017); Camelo & Cretu (2023), where the latter use edge GPUs. For the three power limits, we use $307\,\mathrm{MHz}$, $614\,\mathrm{MHz}$, and $921\,\mathrm{MHz}$ as upper-bound operating GPU frequencies for the device. These are determined based on profiling training of different models and selecting the frequencies that assure a power limit is satisfied irrespective of what model or $b$ is used for the training.

- **Baseline 2:** We select the value of $b$ that minimizes $N_{s_{\mathrm{acc}}}$ on the proxy dataset, but we use the same GPU frequencies as in Baseline 1, so no joint optimization is applied.

- **Fastest configuration:** This baseline serves as an *upper bound* where optimal $f$ and $b$ are jointly selected. Furthermore, this baseline validates the importance of jointly selecting $f$ and $b$, as evidenced by its performance gap compared to other baselines under power constraints. Since the optimal configuration is not known prior to training, we determine it by exhaustively training the given model on the given dataset with all batch sizes to get $N_{s_{\mathrm{acc}}}$, substitute in Eq. (2) and finally select the best $b$ and $f$ configuration on the device.

We repeat every experiment with different five seeds for image classification tasks and three seeds for next character prediction and record the mean and standard deviation.

## 5.2 Training Time Evaluation

Table 2 and Table 3 report the training time comparison of our approach and the three baselines on Nvidia Jetson Nano. Table 4 provides additional comparison for image classification task on Nvidia Jetson TX2NX.

Firstly, we start with the evaluation of image classification tasks reported in Table 2. For ResNet18 training, our method consistently outperforms baseline 1, achieving $1.5 - 2.3\times$ reduction in training time. It also outperforms baseline 2 with $1.4 - 1.7\times$ less training time. These two baselines are only able to explore a small subset of the design space, potentially missing the optimal configuration. Finally, compared to the fastest configuration, our method selects the same configurations in most of the evaluations, while in the case of a different selection, a minor performance decline (of 3%) is observed. Training of MobileNetV2 has three distinct characteristics: lower peak power, better parallelization for larger batch sizes, and different **r**. Our method adapts to that by selecting larger batch sizes (and frequencies) than those selected for ResNet18. Compared with baseline 1, our method performs better with minor differences in training time since both select large batch sizes and maximum $f$ utilizing the power. In low-power, the gain from our approach increases as the selected batch size enables higher frequencies, reducing training time by up to $1.4\times$. Baseline 2 misses the opportunity to use higher frequencies, especially as MobileNetV2 consumes less power.

Table 2: Training time comparison with baselines and upper bound over three different power limits (i.e., $P_{\max}^1$, $P_{\max}^2$, and N/A) on Jetson Nano, showing a peak gain of 2.3×. Recorded times are in seconds. CIFAR10 is used as proxy dataset in our proposed approach. (Evaluation time is excluded)

| Model | Method | SVHN | | | CINIC | | |
|---|---|---|---|---|---|---|---|
| | | $P_{\max}^1$ | $P_{\max}^2$ | N/A | $P_{\max}^1$ | $P_{\max}^2$ | N/A |
| ResNet18 | Baseline 1 | 14347±3796 | 8204±2170 | 6187±1637 | 18154±4109 | 10381±2349 | 7829±1772 |
| ResNet18 | Baseline 2 | 11345±1616 | 7055±1005 | 5817± 828 | 12436±1443 | 7733± 897 | 6377± 740 |
| ResNet18 | Ours | **8477±1207** | **4607± 725** | **4170± 656** | 9292±1078 | **4454± 788** | **4032±713** |
| ResNet18 | Fastest configuration | 8477±1207 | 4607± 725 | 4170± 656 | 9030±1598 | 4454± 788 | 4032± 713 |
| MobileNetV2 | Baseline 1 | 8708±2396 | 5107±1405 | 3990±1098 | 8909±1121 | 5226± 657 | 4082± 513 |
| MobileNetV2 | Baseline 2 | 8082± 912 | 4871± 550 | 3912± 441 | 8428± 692 | 5079± 417 | 4080± 335 |
| MobileNetV2 | Ours | **5940± 916** | **3912±441** | **3912± 441** | **6194± 509** | **4080± 335** | **4080± 335** |
| MobileNetV2 | Fastest configuration | 5940± 916 | 3912± 441 | 3912± 441 | 6194± 509 | 4080± 335 | 4080± 335 |
| EfficientViT | Baseline 1 | 13033±1233 | 7839±741 | 6369±602 | 8100±564 | 4800±339 | 3900±275 |
| EfficientViT | Baseline 2 | 14539±663 | 9269±422 | 7843±357 | 9223±584 | 5880±372 | 4975±315 |
| EfficientViT | Ours | **11036±503** | **6925±655** | 6369±602 | **7001±444** | **4240±299** | 3900±275 |
| EfficientViT | Fastest configuration | 11036±503 | 6925±655 | 6369±602 | 7001±444 | 4240±299 | 3900±275 |

Table 3: Training time comparison with baselines and upper bound for different power limits (i.e., $P_{\max}^1$, $P_{\max}^2$, and N/A). Recorded times are in seconds. Tiny Shakespeare is used as proxy dataset in our approach.

| Model | Method | Austen | | | Dickens | | |
|---|---|---|---|---|---|---|---|
| | | $P_{\max}^1$ | $P_{\max}^2$ | N/A | $P_{\max}^1$ | $P_{\max}^2$ | N/A |
| Transformer | Baseline 1 | 8136±524 | 4607±296 | 3556±229 | 12347±439 | 6991±249 | 5397±192 |
| Transformer | Baseline 2 | 9425±131 | 6081±85 | 5193±72 | 14278±228 | 12941±532 | 7867±125 |
| Transformer | Ours | **7232±101** | **3982±256** | **3556±229** | **10956±175** | **6043±215** | **5397±192** |
| Transformer | Fastest configuration | 7232±101 | 3982±256 | 3556±229 | 10956±175 | 6043±215 | 5397±192 |

Table 4: Training time comparison with baselines and upper bound over three different power limits (i.e., $P_{\max}^1$, $P_{\max}^2$, and N/A) on Nvidia Jetson TX2NX. Recorded times are in seconds. CIFAR10 is used as proxy dataset in our proposed approach. (Evaluation time is excluded)

| Model | Method | SVHN | | | CINIC | | |
|---|---|---|---|---|---|---|---|
| | | $P_{\max}^1$ | $P_{\max}^2$ | N/A | $P_{\max}^1$ | $P_{\max}^2$ | N/A |
| ResNet18 | Baseline 1 | 7343±1943 | 3944±1043 | 3021±799 | 9292±2103 | 4991±1129 | 3823±865 |
| ResNet18 | Baseline 2 | 7337±1943 | 4771±679 | 4086±582 | 8042±933 | 5230± 897 | 4479±520 |
| ResNet18 | Ours | **4283±708** | **2648±417** | **2431±382** | **5463±889** | **2560±453** | **2350±416** |
| ResNet18 | Fastest configuration | 4283±708 | 2648±417 | 2431±382 | 5046±893 | 2510± 625 | 2294±571 |

Ultimately, our method selects the fastest configurations when training MobileNetV2 in all power-limits scenarios. Similarly, our method results in less training times compared to baselines 1 and 2 under $P_{max}^1$ and $P_{max}^2$ when finetuning EfficientViT, while selecting the fastest configuration in all scenarios.

Next, we examine the next character prediction training. As shown in Table 3, our method improves the training time by 1.15× and 2.14× compared to baseline 1 and 2, while choosing the same configurations as the fastest configuration.

In Table 4, we compare our solution with the baselines for ResNet18 training on Nvidia Jetson TX2NX, where the device receives **r** for ResNet18 training from the server. For training on SVHN, our method outperforms both the baselines 1 and 2, reducing the training time by peak of 1.7× and 1.8×, respectively. Furthermore, it selects the same configuration as the optimal one. Similarly, when training on the CINIC dataset, baselines 1 and 2 require up to 1.94× and 2.04× more training time compared to ours. Our method did not select the exact optimal configuration in this scenario; however, it selects a near-optimal configuration that results in only 2% − 8.2% more training time compared to optimal.

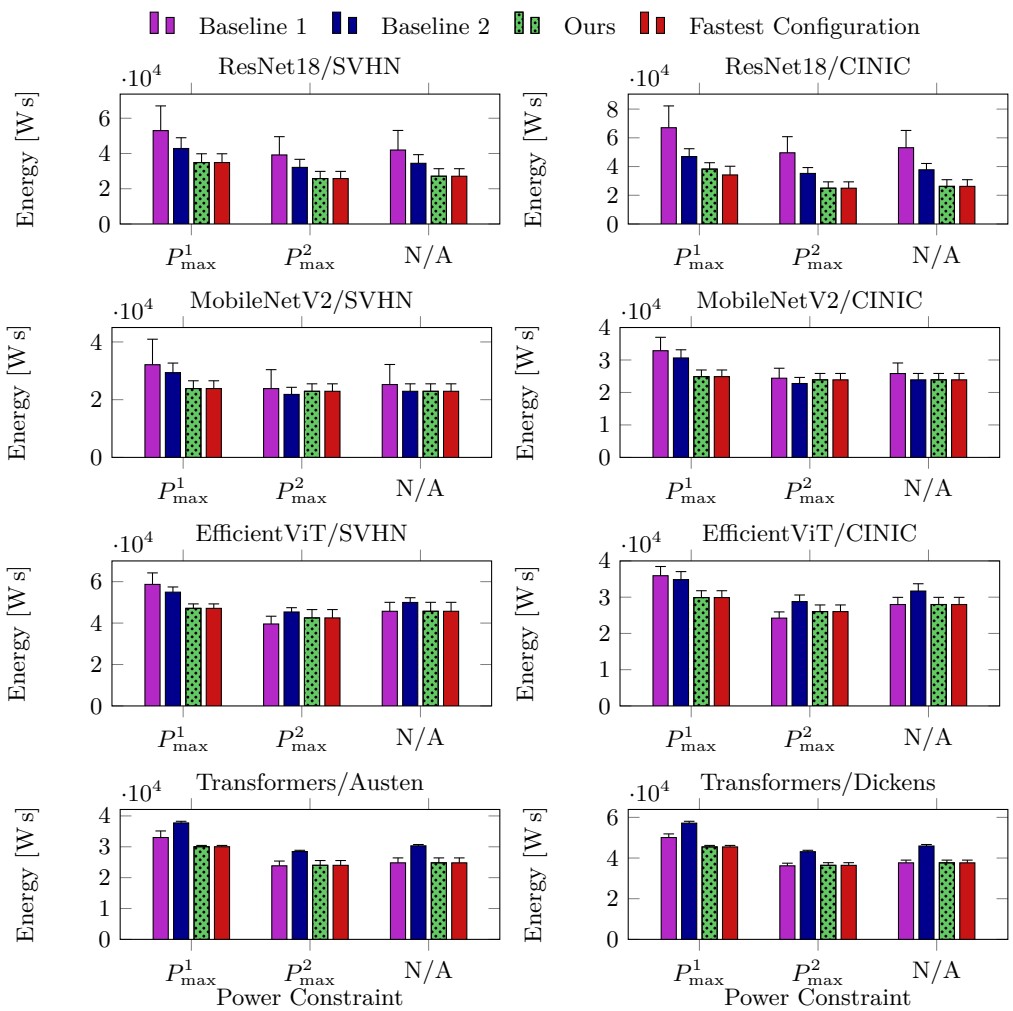

Figure 4: Energy consumption comparison between our approach and baselines during training under three power scenario, using ResNet18, MobileNetV2, and EfficientVIT on the SVHN and CINIC datasets, and a transformers network on Austen and Dickens datasets, performed on Nvidia Jetson Nano.

In summary, our method outperforms existing baselines, reducing training time to accuracy across various model architectures, tasks, and hardware, with an average speedup of 1.5× to baselines 1 and 2 under power constraints $P_{max}^1$ and $P_{max}^2$.

## 5.3 Energy Evaluation

We investigate the energy efficiency of our proposed approach for the image object classification and next character prediction tasks in Figure 4.

For the image classification task, it is noticeable that our method always outperforms baseline 1 since it takes less training time given the power limits. In addition, selections made by baselines 1 and 2 can lead to up to 2× and 1.4× more energy usage compared to our method, respectively. It is important to state that minimizing the training time does not always mean minimizing energy consumption. An example is comparing the energy consumption for $P_{max}^2$ on ResNet18 with the no power limit N/A. Despite having lower training time in N/A (as shown in Table 2), training under $P_{max}^2$ is more energy efficient since the used frequencies for $P_{max}^2$ provide a better tradeoff between performance and power. The same applies for Baseline 2 when training MobileNetV2 and Baseline 1 on EfficientViT under $P_{max}^2$, that uses a lower

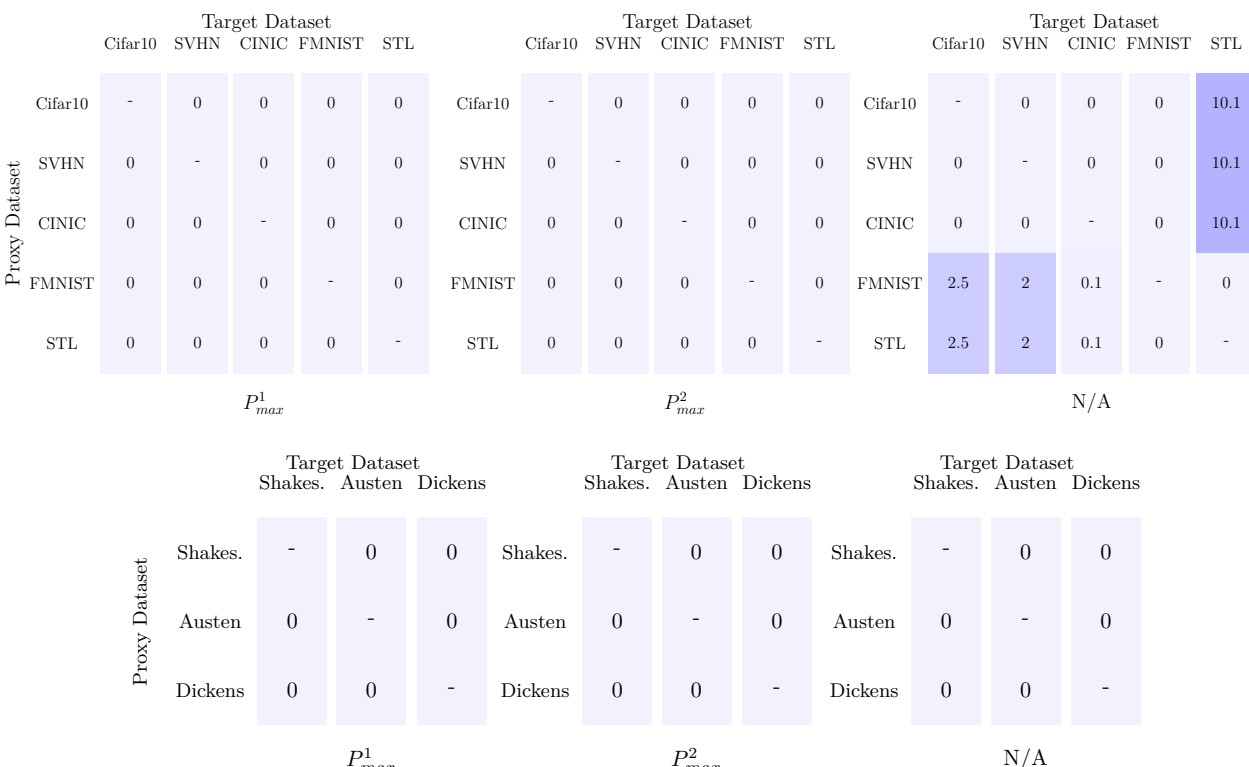

Figure 5: Matrices showing the mean percentage increase in training time of our approach to the fastest configuration for image classification and next-character prediction datasets on Nvidia Jetson Nano across three power constraints. The top matrix corresponds to training MobileNetV2 for image classification tasks, and the bottom to training a transformer network for next-character prediction tasks. Each entry reflect the mean percentage increase in training time when selecting configuration given the proxy-dataset's relation vector **r** (row) instead of the optimal configuration (column) for the target data. A value of 0% denote cases where the same configuration was selected, resulting in no time increase. The results indicate that the proposed method is not sensitive to the selection of proxy dataset, with minimal impact on the training time when the selected configuration differs from the fastest configuration for the target dataset.

frequency with the same batch size of our approach. For the next character prediction task, our approach uses 9% less energy for $P_{\max}^1$ and nearly the same energy consumption for $P_{\max}^2$ while training in less time (the same configuration selection for $P_{\max}^3$) compared to the baseline 1. In addition, our method records $1.25\times$ less training energy for the baseline 2.

## 5.4 Proxy Dataset Analysis

We conduct a sensitivity analysis for the selection of the proxy dataset for image classification and next character prediction tasks by testing all possible proxy and target dataset combinations. We report the mean percentage increase (averaged over runs; see Section 5.1) in training time relative to the fastest configuration.

For the image classification task, we add two additional datasets namely, Fashion MNIST Xiao et al. (2017) (FMNIST) and STL Coates et al. (2011). The tasks include object, digit, and fashion classification, with dataset sizes ranging from $5K$ to $22.5K$ samples. We provide in Fig. 5 a confusion matrices showing the percentage of time increase for the choice of proxy dataset selection compared to the fastest configuration for different datasets for training MobileNetV2 and the transformer network for next character prediction on Jetson Nano. For MobileNetV2, the results show that our approach selects the correct configurations in most of the cases with only a few instances where the optimal configuration is not selected in no power

constraint scenario. The same observation is also applied to ResNet18 as shown in Figure 7 (Appendix A.2). For the next character prediction, the optimal configuration is selected regardless of the selected proxy dataset. To sum up, this analysis shows that our approach is not sensitive to the selection of proxy dataset, demonstrating its practicality.

## 6 Conclusion

In this work, we propose a power-aware training method aimed at accelerating training on power-constrained GPU devices. Our results show that great savings can be achieved in terms of training time and energy consumption, when carefully and jointly selecting the system and application parameters for training, without scarifying the training model quality. The proposed solution is applicable to a wide range of models (including, but not limited to, CNNs and transformers). Our solution can be employed to leverage nearby renewable energy sources by adapting GPU frequency and batch size to the available green energy at edge devices to reduce carbon footprint. A limitation of this work is that we assume that the power constraint on a device does not change through the training period, which may not hold for long-duration tasks that can take multiple days. For future work, we want to extend our study to the varying power constraints and distributed learning settings where devices with different constraints contribute to the training.

## Acknowledgments

This research was supported in part by the EU Horizon project EXIGENCE under Grant Agreement No. 101139120.

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

# A   Appendix

## A.1   Training on edge CPU

In this section, we briefly discuss training on an edge device's CPUs (i.e., Jetson Nano) and the reasons of why to focus only on training on GPUs on such devices. In Fig. 6, we show the processing time and power for training over 50K samples with batch size 32, where both CPU and GPU are using the maximum frequency. We observe that the GPU is an order of magnitude faster compared to the CPU, while nearly consuming similar average power. Similarly, the GPU would be nearly an order of magnitude more energy efficient on these devices compared to the CPU. Thus, we focus in our work on edge GPUs as training is more time and energy efficient.

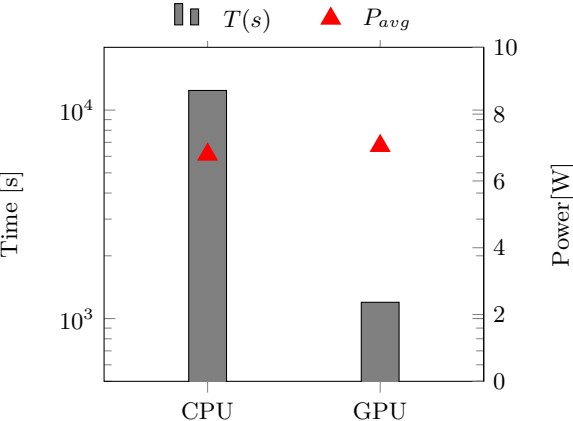

Figure 6: ResNet18 time for training 50K samples ($T_s$) and average power (red triangles) of Cifar10 on Jetson Nano over CPU and GPU.

## A.2   Proxy dataset analysis extension

We provide a continuation for the sensitivity analysis for the proxy dataset selection. In Fig. 7, provide the confusion matrices for selecting different proxy datasets to train ResNet18. The difference for ResNet18 compared to MobileNetV2 (in Figure 5)is that the suboptimal configurations are selected for the lowest power constraint, while for other cases the selected configurations match the optimal.

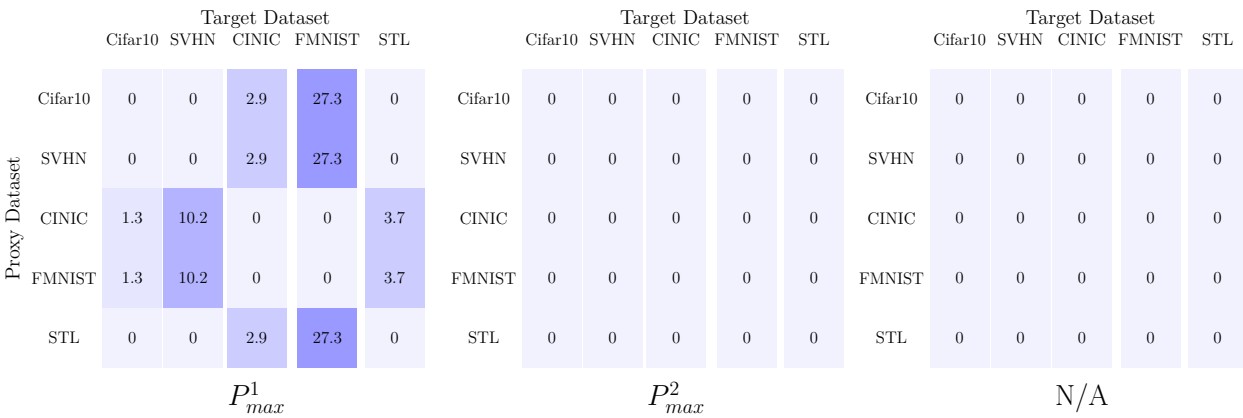

Figure 7: Matrices of mean time increase percentages to the fastest configuration for image classification datasets across three power constraints for ResNet18 training on Jetson Nano.

### A.3 Text datasets details

For Jane Austen's dataset part, we extracted it from the nltk package (gutenberg corpus) where the text consists of 3 novel namely Emma, Persuasion, and Sense and Sensibility. For Dickens, we used eight novels namely The Pickwick Papers, Pictures from Italy, A Tale of Two Cities, A Story of the French Revolution, The Chimes, Mugby Junction, The Haunted Man and the Ghost's Bargain, and The Mystery of Edwin Drood. We filter the licensing text, author history, etc. from the training and testing text.

### A.4 Configuration selection visualization

In this section, we provide additional visualization for our method. Figures 8 and 9 illustrate the configuration candidates from our method under power constraints, together with the cases of no joint optimization. Specifically, this includes the power constraint cases of $P^1_{max}$ and $P^2_{max}$ for ResNet18 in Figure 8 and transformers for the next character prediction in Figure 9. Our approach finds a wider range of operating points in the search space, effectively adapting to different training workloads. For instance, the next-character prediction task is less power-intensive, which allows exploring a broader set of operating points. Additionally, it selects the best configuration to minimize $TT_{acc}$.

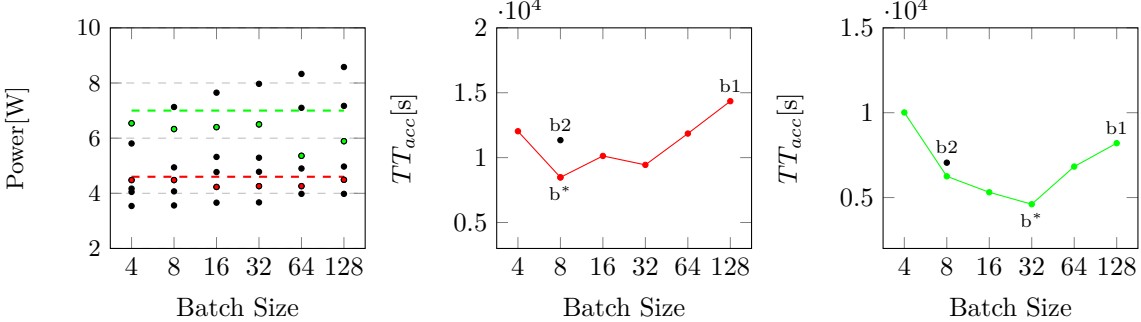

Figure 8: Visualization of the the power and resulting training times to accuracy for the configuration candidates of our method when finetuning ResNet18 on SVHN under power constraints of $P^1_{max}$ (red) and $P^2_{max}$ (green) respectively. We highlight baselines 1 and 2 (b1, b2), corresponding to the cases without joint optimization, together with b*, denoting the point selected by our method.
. Our method explores more operating points in the search space while selecting the best configuration.

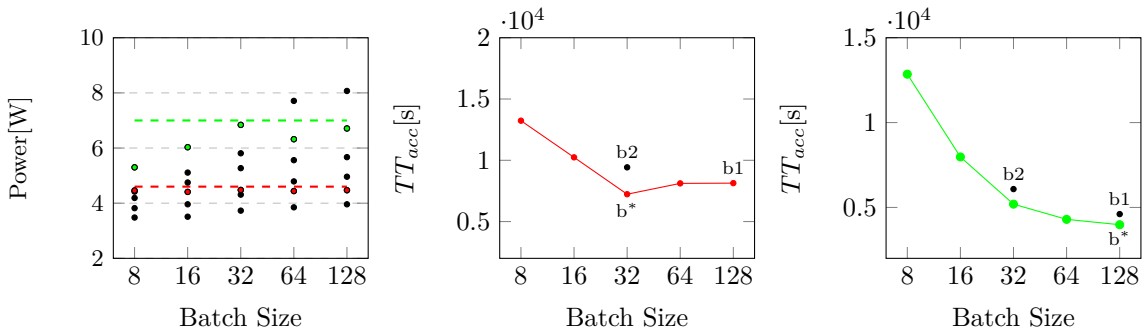

Figure 9: Visualization of the the power and resulting training times to accuracy for the configuration candidates of our method when finetuning transformer on Austen dataset under power constraints of $P^1_{max}$ (red) and $P^2_{max}$ (green). We highlight baselines 1 and 2 (b1, b2), corresponding to the cases without joint optimization, together with b*, denoting the point selected by our method. Our method explores more operating points in the search space while selecting the best configuration.

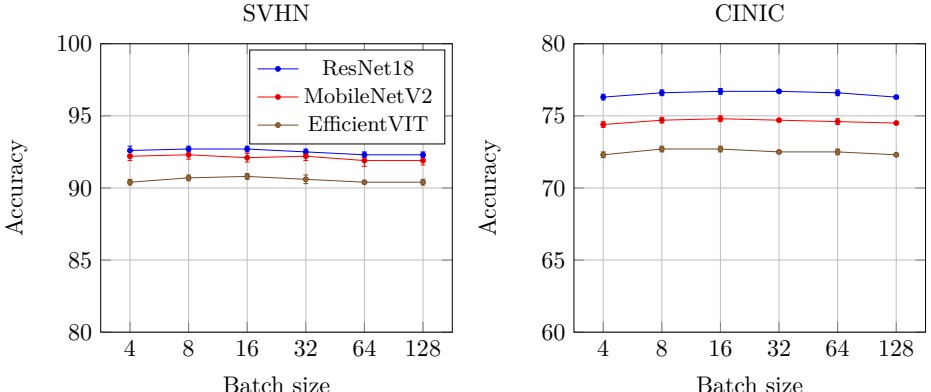

Figure 10: Maximum test accuracy across different batch sizes for the image classification task, showing stable performance with only marginal variation.

### A.5 Effect of batch Size on model quality

To complement our main results, we analyze the impact of batch size on training quality. Specifically, we train the image classification models under different batch sizes and show the mean of maximum accuracy achieved over three runs on test set in Fig. 10. The max accuracy remained consistent across different batch sizes, showing only marginal differences without signs of increased overfitting. Therefore, the proposed configuration selection approach can select the batch size and frequency combination that improves the training speed given a device's power constraint and capabilities without compromising model quality. This is consistent with prior work Goyal et al. (2017) showing that larger batch sizes can be used without loss in accuracy.

### A.6 Power measurement details

We read the power sensor values during training with a sample rate of 1s. For the Nvidia Jetson Nano, the power measurements are available to read from using sysfs nodes. For example, on the jetson nano, the sysfs nodes are available in the path '/sys/bus/i2c/drivers/ina3221x/6-0040/iio:device0/'.

