# OpenReview forum: "Accelerated Training on Low-Power Edge Devices"
_TMLR — Accepted by TMLR_

### Review · Reviewer_rhhh · 2025-08-04

**Summary Of Contributions:**

This work focuses on addressing the challenge of efficient deep learning model training on power-constrained edge devices.

Its key contribution is a cross-layer methodology that jointly optimizes system parameters (GPU frequency) and application parameters (batch size). This approach, which combines offline device profiling with server-side batch size efficiency estimation, dynamically selects the best configuration to minimize training time while adhering to power limits.

Experiments show this method reduces training time by up to 2.3x and lowers energy consumption compared to baselines, without sacrificing model performance.

**Audience:**

Yes

**Audience Explanation:**

Training on power-constrained edge devices is a relevant topic for the community. Jointly optimizing GPU frequency and batch size, can be a topic that benefits sustainable ML.

**Claims And Evidence:**

Yes

**Claims Explanation:**

The paper is well written and is relatively easy to understand. The authors provided performance evaluations to support their claim. However, I have a few questions:

- Since this method utilizes an estimation method for deciding the batch size. It is surprising to see in Tables 2 and Table 3 that the performance of the Joint optimization is actually as good as the optimal one. Is there any case where the proposed method will not give us the optimal?

- Even though it is beneficial for training speed to jointly optimize both batch size and GPU freqency, more analysis should be provided in terms of the other effects of using different batch sizes, such as max accuracy achievable, generalization of the model, overfitting, etc.

- It is unclear how to select a proxy dataset in a real training scenario, and how the proxy dataset is used to estimate the real dataset for server-side batch size efficiency estimation is not too clear.

**Requested Changes:**

- For figure 8 and figure 9, it would be beneficial to also show the curves without join optimization and just optimize for the best batch size, so the readers can better understand the benefit of the join optimization.

- address the concerns mentioned above

---

> ### Author Response · Authors · 2025-09-04
>
> We thank the reviewer for the review and constructive feedback.
>
> ### Is there any case where the proposed method will not be the fastest ?
>
> - Yes, such cases can occur. For instance, as shown in Table 2 (ResNet for CINIC dataset under $P^{1}_{max}$ ) and figure 5 (the N\A scenario) , the proposed approach selects a near-optimal configuration compared to the fastest one. This is primarily due to a small error in batch size estimation, which can lead to selecting operating points on the device that are near, but not identical to, the optimal configuration, especially when there exist multiple operating points yielding similar performance.
>
>
> ###  More analysis on the other effects of using different batch sizes, such as max accuracy achievable, generalization of the model, overfitting, etc.
>
> - Regarding model quality, we conducted an analysis (in Appendix A.5) on image classification and observed that the max accuracy and generalization performance remained consistent across the different batch sizes, showing only marginal differences, without signs of increased overfitting. We further refer to [R1], which showed that with learning rate scaling, as also adopted in our setup, it is possible to train with large batch sizes (of up to 8k) without loss in accuracy.
>
> [R1] Goyal, Priya, et al. "Accurate, large minibatch sgd: Training imagenet in 1 hour." arXiv preprint arXiv:1706.02677 (2017).
>
> ### Further clarification on selection of a proxy dataset in a real training scenario and the server-side efficiency estimation.
>
> - The proxy dataset should correspond to the same task type (e.g., character prediction, image classification). As demonstrated in our experiments (section 5.4), using a single proxy dataset provides good estimate of the configuration on the device. This makes the approach practical in real scenarios, since once the proxy dataset is used, the same estimated relation can reliably guide the configuration selection on the device for any other dataset of the same task type, without requiring repeated proxy selection or additional overhead. Furthermore, we further clarified the estimation process in section 4.2 by stating that each batch size run starts from the same pre-trained weights and is trained on the proxy dataset until convergence to obtain its $N_s$, from which the batch size relation vector is derived.
>
> ### Suggestion of adding without joint optimization cases in figures 8-9 for the baselines.
> - Thank you for the suggestion. We added the requested changes in the figures.

---

### Review · Reviewer_vngf · 2025-08-18

**Summary Of Contributions:**

This paper introduces a novel cross-layer methodology to accelerate the training of deep learning models on power-constrained edge devices. The authors argue that conventional methods, which typically reduce GPU frequency to meet power limits, lead to significantly longer training times. Their proposed solution is to jointly optimize the GPU frequency (a system parameter) and the training batch size (an application parameter). The methodology involves two main phases: offline profiling of the neural network's power consumption and training time on the target device for various frequency and batch size combinations, and estimating the relationship between batch size and the number of iterations required to reach a target accuracy. This estimation is done on a powerful server using a proxy dataset. By combining these two pieces of information, the system can select the optimal frequency-batch size pair that minimizes the total training time while respecting the device's power constraints.

1. It clearly shows that independently setting the GPU frequency and batch size is suboptimal for training on power-constrained edge devices.

2. The authors propose a practical method that combines device-specific hardware profiling with server-based training analysis (using a proxy dataset) to find the best operating point (batch size and GPU frequency).

3. The evaluation on real hardware (Nvidia Jetson Nano and TX2NX) demonstrates a reduction in training time by up to 2.3x and an average of 1.5x compared to baseline approaches, without compromising the final model's performance.

4. The method is validated across various models (ResNet18, MobileNet V2, EfficientViT, Transformers), tasks (image classification, next character prediction), and datasets, showing its robustness and practicality. The authors also analyze the impact on total energy consumption, noting significant reductions.

**Audience:**

Yes

**Audience Explanation:**

Low-power ML should be in the scope of interests of TMLR.

**Claims And Evidence:**

Yes

**Claims Explanation:**

This version of paper has been improved a lot. The authors have revised their performance claims and corrected data in the results tables, making the latest version more accurate and trustworthy.

This version provides a much more detailed and specific summary of the contributions, giving the reader a clearer understanding of the work's scope from the outset. The contributions are explicitly detailed, mentioning the exact models and tasks: "as shown across representative models (CNNs and Transformers) and tasks (image classification and next character prediction)". It also explicitly names the models and datasets used in the evaluation summary bullet point. This level of detail is a significant improvement.

By correcting the main performance metric (2.4x=>2.3x) and a key data point in the results, the authors have substantially improved the paper's trustworthiness.

**Requested Changes:**

I don't have further requests here.

---

> ### Author Response · Authors · 2025-09-04
>
> We thank the reviewer for their review and constructive feedback.

---

### Review · Reviewer_PcJG · 2025-08-25

**Summary Of Contributions:**

This paper proposes a new method to accelerate neural network training on edge devices under power constraints by jointly optimizing GPU operating frequency and training batch size. Traditionally, only GPU frequency was limited to meet power constraints, which caused a large increase in training time. This study introduces a cross-layer control method spanning the hardware layer (GPU frequency) and the application layer (batch size), and accelerates training by adjusting both simultaneously within the device's power cap. Specifically, it predicts the efficiency of each batch size (the number of iterations needed to reach a target accuracy) using a proxy dataset, and profiles processing time and power for each batch size-frequency pair on the device, then selects the combination that minimizes total time to the target accuracy. As a result, the proposed method achieves up to a 2.3x reduction in training time (1.5x on average) over current baselines, and shows on real devices that total energy consumption is also reduced. It also achieves these improvements without degrading the final model accuracy. In addition, sensitivity analyses on five image classification datasets and three character-level prediction datasets verify the robustness and practicality of the method with respect to proxy data selection.

Main strengths: The paper addresses an important problem of on-device learning on edge devices. The proposed method is an interesting approach that integrates hardware control and machine learning parameter tuning, and achieves large performance gains through the joint optimization of GPU frequency and batch size, which had not been explored before. Experiments are conducted comprehensively on real low-power GPU devices such as Jetson Nano and TX2NX, and the effectiveness is shown across multiple models and tasks, including image classification (ResNet18, MobileNetV2, EfficientViT) and natural language (character-level prediction with a Transformer), suggesting good generality. It also considers practically important points such as reducing energy consumption and maintaining accuracy, supporting the effectiveness and usefulness of the proposal.

Main weaknesses: On the other hand, the novelty is a relatively simple optimization idea (combining elements that had been tuned separately in existing techniques), so it remains a systems-side engineering method rather than a theoretical breakthrough in machine learning algorithms. In other words, while the contribution is important for applications, the academic innovation is somewhat incremental. In addition, the method assumes that the device's power budget remains constant during training, and it cannot directly handle cases where the battery level changes over long training or where the power constraint itself changes. Furthermore, because estimating the optimal batch size requires pretraining with a proxy dataset, the selection of that dataset and the additional cost are practical issues in real deployments.

**Audience:**

Yes

**Audience Explanation:**

There will be readers within TMLR's audience who are interested in this work. In recent years, machine learning on edge devices (on-device learning) and energy-efficient AI have become highly active research areas. This study fits exactly in that area and addresses a practical and somewhat urgent problem of improving training efficiency under resource constraints; the results (significantly shorter training time and improved energy efficiency) will be useful information for researchers and engineers interested in edge AI and embedded machine learning.

**Broader Impact Concerns:**

There are no particular serious ethical or social concerns for this study. Rather, by improving the efficiency of on-device learning, the method may contribute to privacy protection (training finished on the device without sending data to the cloud) and to power savings, so positive impacts are expected.

**Claims And Evidence:**

Yes

**Claims Explanation:**

The claims in the paper are supported by sufficiently convincing evidence. The authors claim that the proposed method "speeds up training by 1.5x on average" and "reduces energy consumption," and this is backed by detailed experiments on real devices. For example, on Jetson Nano with ResNet18 for image classification, the method reaches the target accuracy 1.5x faster on average than existing methods (Baseline 1: frequency limiting under a power cap with maximum batch size; Baseline 2: selecting a batch size that converges faster, etc.), with a peak speedup of 2.3x. These results are clearly shown in tables and graphs. Consistent improvements are also observed for other models and tasks such as MobileNet and Transformer, indicating effectiveness not limited to specific cases. For energy consumption, the proposed method uses less than the baselines, and in some cases it is reported to require about half the energy of prior methods. Therefore, the paper's claims are clearly supported by abundant experimental results and analysis, and the evidence is reliable.

**Requested Changes:**

Please further clarify the positioning of the proposed method and the differences from prior work in the introduction and related work. In particular, compared with existing methods for similar problems (e.g., those that control only GPU frequency or that adjust batch size for inference), emphasizing what this work achieves for the first time will make its novelty clearer to readers. From the current text it can be inferred that "joint optimization of batch size and frequency has no precedent," but based on the early example that shows a 31.9% speedup in configurations that prior methods could not explore, please explicitly summarize the uniqueness of your method.

---

> ### Author Response · Authors · 2025-09-04
>
> We thank the reviewer for their review and constructive feedback.
>
> In response to the requested change of further clarifying the uniqueness of our method, we have revised the introduction to highlight the positioning and uniqueness of the proposed approach and its distinction from optimizing batch size in inference.

---

### Comment · Action_Editor_22kx · 2025-08-26
**Start author-reviewer discussions**

Dear authors,

You have three reviews. Please address their comments as soon as possible. The discussion period will end on September 8th, 2025.

---

Dear reviewers,

Thank you for submitting your reviews. As this paper received three reviews, now it's time to have discussions with authors.

You will be able to submit your formal decision recommendation starting in 2 weeks. Your prompt responses to authors' comments would be very appreciated.

Remember that different from other journals / conferences, [TMLR's acceptance criteria](https://jmlr.org/tmlr/reviewer-guide.html) are based on positive answers to the following two questions

- Are the claims made in the submission supported by accurate, convincing and clear evidence?
- Would some individuals in TMLR's audience be interested in the findings of this paper?

Some of you mentioned novelty of the method, but novelty of the studied method is not a necessary criteria for acceptance at TMLR.

It's still not too late to review [the TMLR's reviewer guideline](https://jmlr.org/tmlr/reviewer-guide.html) in case you are not aware of that.


Thank you!
Your AE

---

> ### Comment · Action_Editor_22kx · 2025-09-09
> **Your official recommendation is due September 22th**
>
> Dear Reviewers,
>
> Now it's time for you to submit an official recommendation for this submission.
>
> The authors responded to your initial reviews several days before the end of the discussion period.
> I have not seen any responses to the author rebuttal yet.
>
> Please
> 1. consider their rebuttal comments,
> 2. respond to the comments ***ASAP*** if you have follow-up questions or comments, and
> 3. submit your official recommendation ***by September 22th***.
>
> In your official recommendation, please share your thoughts on the author rebuttal, specifically, what point in your review their responses (did not) address. It will greatly help me write a meta-review and make a fair decision.
>
> Thank you,
> Your AE

---

### Decision · Action_Editor_22kx · 2025-09-29

**Recommendation:** Accept as is

**Additional Comments:**

Two reviewers suggested acceptance and one reviewer was leaning towards rejection without justification. I sent the reviewer a few reminders and asked for clarifications, but didn't receive any response.
The only potential negative points I found in their review are not as critical as [TMLR's acceptance criteria](https://jmlr.org/tmlr/acceptance-criteria.html).
Besides, the three reviewers confirmed that key claims in this paper are supported by evidence and they did not request additional revisions.

For these reasons, I recommend TMLR accepts this paper as is.

**Audience:**

Yes

**Audience Explanation:**

Since the key contents of this revision is the same as the previous submission, I refer to my previous answer:
> Training on low-power edge devices should be of interest to this community.

**Claims And Evidence:**

Yes

**Claims Explanation:**

This is my second time to assess this work as a TMLR AE. This revision includes corrected claims that are better aligned with their experimental results than those in the previous submission.

Some of the previous reviewers reviewed this revision and confirmed that it's more accurate and trustworthy. All reviewers suggest that the claims in the manuscript are supported by the evidence.